# Transcribed Ultraconserved Regions in Cancer

**DOI:** 10.3390/cells11101684

**Published:** 2022-05-19

**Authors:** Myron K. Gibert, Aditya Sarkar, Bilhan Chagari, Christian Roig-Laboy, Shekhar Saha, Sylwia Bednarek, Benjamin Kefas, Farina Hanif, Kadie Hudson, Collin Dube, Ying Zhang, Roger Abounader

**Affiliations:** 1Department of Microbiology, Immunology, and Cancer Biology, School of Medicine, University of Virginia, Charlottesville, VA 22908, USA; mkg7x@virginia.edu (M.K.G.J.); as7fr@virginia.edu (A.S.); bc8he@virginia.edu (B.C.); rtf6mq@virginia.edu (C.R.-L.); ss7st@virginia.edu (S.S.); gdn9px@virginia.edu (S.B.); bak4x@hscmail.mcc.virginia.edu (B.K.); pzu3ps@virginia.edu (F.H.); rce3ka@virginia.edu (K.H.); cjd7ua@virginia.edu (C.D.); yz5h@virginia.edu (Y.Z.); 2Department of Neurology, School of Medicine, University of Virginia, Charlottesville, VA 22908, USA; 3NCI Designated Comprehensive Cancer Center, University of Virginia, Charlottesville, VA 22908, USA

**Keywords:** TUCR, ultraconserved, cancer, long non-coding RNA

## Abstract

Transcribed ultraconserved regions are putative lncRNA molecules that are transcribed from DNA that is 100% conserved in human, mouse, and rat genomes. This is notable, as lncRNAs are typically poorly conserved. TUCRs remain very understudied in many diseases, including cancer. In this review, we summarize the current literature on TUCRs in cancer with respect to expression deregulation, functional roles, mechanisms of action, and clinical perspectives.

## 1. Introduction

Much of the molecular research on cancer is focused on pathways that are predominantly represented by proteins and protein coding genes. While 90% of the genome is transcribed, only 2% is translated into functional proteins and peptides. The remaining 98% is made up of various non-coding RNA molecules. These non-coding RNAs are commonly annotated by size. Long non-coding RNAs (lncRNAs) are transcripts that are ≥200 nucleotides (nt) in length. Small non-coding RNAs (sRNAs) are all those <200 nt in length. Antisense and enhancer RNAs are subclasses of non-coding RNAs across both size distinctions that are involved in gene regulation. Antisense RNAs typically contain sequences that are complementary to another gene and commonly downregulate genes. Enhancer RNAs are transcribed from enhancer regions in the genome and can have broad regulatory effects [1,2,3,4,5,6,7,8].

Non-coding RNAs were previously often overlooked regarding cancer pathogenesis due to their assumed non-coding nature. However, more recent research demonstrates their broad and deep involvement in the development and progression of most cancers [1,2,3,4,5,6]. The roles for many lncRNAs in cancers have been characterized and strategies to target or use them for therapeutic purposes have been developed and tested in vitro and in vivo [1,2,3,4,5,6]. The findings point to important regulatory roles of lncRNAs in cancer biology and to their potential as prospective novel therapeutic targets.

One relatively understudied subclass of lncRNAs was described in 2004 by Bejerano et al. [7] and is transcribed from ultraconserved regions (UCRs) in the genome. These transcribed ultraconserved regions (TUCRs) are 481 transcripts of ≥200 nt in length that are 100% conserved in human, mouse, and rat genomes. For example, uc.8 is 100% conserved from human to elephant and 83% conserved from human to zebrafish. A comparatively sized lncRNA, LINC02079, is only 40% conserved from human to elephant and 23% conserved from human to zebrafish (Figure 1). TUCR conservation is noteworthy, as lncRNAs as a class are typically relatively poorly conserved (30–40%), and evolutionary conservation is often considered a marker for biological significance.

TUCRs can be found on all numbered chromosomes and the X chromosome (Figure 2). They are non-randomly distributed in the human genome, possibly due to strong negative selection [7]. This is remarkable, as such a high degree of conservation is not necessary to maintain singular functions, such as protein coding, nucleic acid binding, or nucleic acid–protein interactions. Instead, it was suggested that these regions are highly conserved at their genomic loci because they may have several functional roles. For example, a coding TUCR within an exon may also bind splicing factors [7].

TUCR nomenclature consists of “uc.” (ultraconserved) and a number that reflects their location on the chromosomes starting with chromosome 1 and ending with the X chromosome. For example, uc.1 is found on chromosome 1, and uc. 481 is found on the X chromosome (chromosome 23). Sometimes, TUCRs are also referred to by strand. For example, uc.416+ is transcribed by the plus strand DNA and uc.83- is transcribed by the minus strand DNA.

Dysregulation of TUCRs is associated with cancer [1,8,9,10,11,12,13,14] as well as several human diseases that include neurological, cardiovascular, and developmental conditions [12,15,16,17,18,19,20,21]. Reversal of TUCR dysregulations may constitute promising cancer therapies [10]. However, TUCRs remain an understudied class of molecules. As of this review, a PubMed literature search reveals 69 papers that contain the words TUCR, UCR, or ultraconserved and the word cancer in the abstract. In comparison, a PubMed literature search of other classes of non-coding RNA, or even a single gene, such as TP53 (p53), and cancer yielded thousands of publications (Table 1). Therefore, the study of TUCRs can represent novel and impactful research, as there are many questions about their expressions, functions, mechanisms of action, and potential therapeutic exploitation that remain to be answered.

In this review, we summarize published research on TUCR roles in cancer. First, we explore studies that investigated TUCR expression and deregulation in cancer. Then, we discuss various cellular and molecular mechanisms that are affected by TUCR dysregulation. Then, we highlight TUCRs that are associated with clinical parameters. Lastly, we summarize important caveats and controversies to consider when investigating TUCRs, and discuss some questions that remain to be answered.

## 2. Expression and Deregulation of TUCRs in Cancer

TUCRs are found in various genomic loci. They can be intragenic (within an annotated genomic region) or intergenic (outside of current genomic annotations) (Figure 3). Intragenic TUCRs can be located within the exons (exonic) or introns (intronic) of their “host” genes, or spanning exons and introns (exonic/intronic) [22].

The expression of many TUCRs is affected by CpG island hypermethylation, which induces silencing of tumor suppressive TUCRs. Oncogenic TUCRs are reactivated in cancer, perhaps through the reversal of hypermethylation. TUCRs are also significantly enriched for the H3K4me3 marker of active transcription. Even intergenic TUCRs, those that exist outside of annotated genomic regions, exhibit expressions that are significantly higher than noise. This is despite the fact that these intergenic TUCRs are commonly located in genomic deserts, or regions that extend more than 1 megabase (Mb) [23]. While TUCRs are highly conserved—there is significant interspecies and within-species variability in the degree of methylation of high CpG density sequences. Methylation is variable within individual TUCRs but conserved between related species. It is heritable through speciation while experiencing relatively lower selection pressure compared to sequences themselves [24].

TUCRs are broadly deregulated in cancer and are commonly associated with the deregulation of DNA replication and other aspects of the cell cycle. They have been analyzed for expression deregulation in chronic lymphocytic leukemia; colorectal carcinoma; hepatocellular carcinoma; neuroblastoma; colon, bladder, breast, pancreatic, prostate, and lung cancer; melanoma; leukemia; and lymphoma. In many of these cancers, the overexpression of potential oncogenic TUCRs and the downregulation of tumor suppressive TUCRs is associated with increased cancer risk [25].

Deletion or duplication of an ultraconserved region (UCR) can be deleterious to the mammalian cell. Copy number variants (CNVs) that arise somatically and are relatively newly formed are less likely to have established a CNV profile that is depleted for UCRs. It is possible to observe the CNVs of induced pluripotent stem (iPS) cells become depleted of UCRs over time, suggesting that depletion may be established through selection against UCR-disrupting CNVs without the requirement for meiotic divisions. Because of their ultraconservation and overall resistance to variation in healthy cells, they represent potential controls in studies analyzing the overall mutability of neighboring genes [26,27,28].

UCRs, similar to microRNAs, are frequently located at fragile sites and genomic regions that are affected in various cancers, which are named cancer-associated genomic regions (CAGRs). This observation appears counterintuitive, as UCRs are ultraconserved despite being enriched in regions that are susceptible to alterations. There is a need for methods to detect cancer-associated genomic instability that do not require already established neoplastic cells to detect mutations, as early detection is crucial to treatment. Ultraconserved regions accumulate fewer mutations than surrounding segments in samples from neoplastic and non-neoplastic hereditary non-polyposis colorectal cancer patients. As no difference was found in healthy donors, this indicates that UCRs may be an effective internal control for early detection of genomic instability [29,30].

It has been speculated that ultraconserved regions in non-coding DNA have been conserved through weak negative selection. However, analysis of the derived allele frequency shows that these regions are under much stronger negative selection than those of protein-coding genes [31]. Although TUCRs themselves are significantly more resistant to variation than these less conserved genomic regions, single nucleotide polymorphisms (SNPs) within TUCRs are associated with increased cancer risk [32]. Several studies have suggested the use of TUCRs in conjunction with or independent of their associated SNPs as potential prognostic markers at the genomic level [31].

The distribution of fitness effects of new mutations in ultraconserved regions, inferred from within-species polymorphism, suggests that ultraconserved regions receive more strongly selected deleterious mutations and fewer nearly neutral mutations than amino acid sites of protein-coding genes or regulatory elements close to genes. However, it was also shown that ultraconserved regions experience a much greater proportion of adaptive substitutions than any known category of genomic sites in murids. These findings suggest that there is widespread adaptation in mammalian conserved non-coding DNA elements, some of which have been implicated in the regulation of crucially important processes, such as development [33].

Previous studies have shown that ultraconserved regions (UCRs) can act as enhancers in mice. Studies show that the concurrent presence of enhancer and transcript function in non-exonic UCR elements is more widespread than previously thought. Additionally, RNAs encoded by non-exonic UCRs are likely to be long RNAs transcribed from only one DNA strand. The transcription products of these UCRs may operate as enhancer RNAs [34]. However, deletion of UCRs that function as enhancers in mice and that are near genes that exhibit phenotypes with altered expression results in viable and fertile mice lacking abnormalities in a variety of phenotypes. Alteration of genes close to UCRs also reveals no change in phenotype. Therefore, UCRs may not always serve functions that are essential to animal viability [35].

Analysis of 14 SNPs within UCRs in prostate cancer patients reveals that the intron variant rs8004379 is associated with recurrence of localized disease as well as a decreased risk for prostate cancer–specific mortality. In silico studies suggest that the SNP also affects NPAS3 expression, which is correlated with patient prognosis. This highlights the potential for SNPs within UCRs to serve as biomarkers of prognosis [36,37]. In colorectal cancer (CRC), eight single nucleotide polymorphisms associated with different stages of the disease were identified within UCRs. Patients with stage II CRC who had at least one variant of the reference SNP sequence 7849 (rs7849, 3′ UTR variant) had increased recurrence risk. Other SNPs which were significant in the training set, but not the validation set, were found. These included rs2421099, rs16983007, and rs10211390 for recurrence and rs6590611 for survival in stage II CRC. SNPs rs6124509 and rs11195893 were also associated with increased recurrence in patients with stage III CRC [38,39]. The functional implications of these SNPs remain unclear.

Distinct profiles of TUCR dysregulation are associated with the conversion of native esophageal epithelium into Barrett’s esophagus and with the progression of Barrett’s esophagus into Barrett-related adenocarcinoma. These include upregulation of uc.58, uc.202, uc.207, and uc.223 and downregulation of uc.214. Barrett’s esophagus was found to consistently show downregulation of uc.161, uc.165, and uc.327 and upregulation of uc.153, uc.158, uc.206, uc.274, uc.472, and uc.473 with analogous profiles in human and murine samples [40]. In neuroblastoma, studies have shown correlations between specific TUCR expression levels and important clinicogenetic parameters such as MYCN amplification status. TUCR relationships to cellular processes such as TP53 response, differentiation, and proliferation were verified using various cellular model systems. Thus, TUCRs are involved in diverse cellular processes that are deregulated in the process of tumorigenesis [22].

## 3. Functions and Mechanisms of Action of TUCRs in Cancer

TUCRs have diverse impacts on cancer initiation and progression through several mechanisms of action. Several molecular pathways are affected by TUCR expression, either directly or indirectly. TUCRs commonly regulate and are regulated by miRNAs [41]. Interplay between TUCRs and other lncRNAs and miRNAs, and the importance of such interactions during the tumorigenic process, provides new insight into the regulatory mechanisms underlying several ncRNA classes of importance in cancer [8].

Several primarily nuclear TUCRs that are upregulated by hypoxic conditions characteristic of the tumor microenvironment are upregulated in colon cancer. Hypoxia-inducible factor appears to be partially responsible for the induction of many of these TUCRs. Uc.475 is intronic within O-linked N-acetylglucosamine transferase which is upregulated in epithelial cancer types and supports proliferation under hypoxic conditions [42].

In breast, prostate, and bladder cancer, uc.63 is upregulated and associated with poor prognosis. Downregulation of uc.63 via genetic silencing leads to reduced cell proliferation. Overexpression leads to increased cell accumulation and migration. It increases the expression of MMP2 through the regulation of miR-130b [43,44,45]. In colon cancer, the transcript containing uc.138 is upregulated. This upregulation is associated with malignant colon cancer. Overexpression of this transcript leads to increased cell cycle progression and resistance to apoptosis [46]. In non-small cell lung, bladder, and prostate cancers, uc.454 is downregulated. Overexpression of uc.454 results in inhibited cell proliferation and induction of apoptosis. These effects could be the result of interactions with the heat shock protein family A member 12B (HSPA12B) or Ras signaling pathway-related transcripts like RIN2 and RAB37 [1,47,48]. In renal cell carcinoma, uc.416 is upregulated and regulated by miR-153. Inhibition of uc.416 reduced cell growth and migration. These effects could be the result of epithelial to mesenchymal transition via downregulation of e-cadherin and upregulation of snail and vimentin. In gastric cancer, uc.416 is upregulated and regulated by miR-153 at an identified binding site. Uc.416 also affects cell growth in gastric cancer via its regulation of IGFBP6, indicating that uc.416 may be part of an oncogenic pathway in the disease [49]. In breast cancer, uc.38 operates as a tumor suppressor by inhibiting cell proliferation and inducing cell apoptosis. This is likely mediated by the regulation BCL-2 family proteins through PBX-1 depletion [50]. In esophageal squamous cell carcinoma, the expression of uc.189 is significantly upregulated. Additionally, high levels of uc.189 expression is associated with invasion of the tumor, advanced clinical stage, lymph node metastasis, and poor prognosis [51,52]. In lung and colorectal cancer, uc.339 is upregulated and believed to act as a competing target RNA with cyclin E2 for miR-339-3p, -663b-3p, and 95-5p. This, in combination with p53 loss can lead to centrosome hyperamplifications. The full RNA transcript encoding uc.339, TUC339, was functionally implicated in modulating tumor cell growth and adhesion. Suppression of TUC339 by siRNA reduced HCC cell proliferation, clonogenic growth, and growth in soft agar. Thus, intercellular transfer of TUC339 represents a unique signaling mechanism by which tumor cells can promote HCC growth and spread [1,53,54,55].

The lncRNA that is transcribed from uc.283 has been shown to prevent pri-miRNA cleavage by Drosha. Mutation of the site in either RNA molecule uncouples regulation in vivo and in vitro. Uc.283 impairs microprocessor recognition and efficient pri-miRNA cropping [47,56]. Uc.158 expression is upregulated in Wnt/β-catenin-dependent hepatocellular carcinoma relatively to non-dependent carcinoma and normal liver cells, and appears to be activated by the Wnt pathway in liver cancer. It is negatively regulated by miR-193b. In addition, inhibition of uc.158 reduced cell growth, spheroid formation and migration, and increased apoptosis [57]. In cervical cancer, uc.206 is significantly upregulated and negatively correlates with the expression of p53 by targeting the 3′ untranslated region of p53. This occurs even though p53 is frequently degraded in cervical cancer due to HPV-mediated decay. Thus, uc.206 acts as a novel oncogene by targeting p53 and enhancing cervical cancer cell line growth. However, further research is needed to fully decipher its role in tumor biology [58]. In lung cancer, uc.83 is upregulated. It affects the PI3K and MAPK pathways via AKT and ERK downregulation, respectively [59]. In breast cancer, uc.183, uc.110, and uc.84 are negatively co-regulated with miR-221 and are involved in the control of CDKN1B expression. Treatment with anticancer drugs that inhibit the cell cycle increased expression of these TUCRs while leaving miR-221 unaffected [60].

The functions and mechanisms of action of TUCR in the cancers described above highlight the variety and breadth of functions and mechanisms of actions of TUCR in the regulation of cancer biology.

## 4. Translational and Clinical Perspectives Associated with TUCRs

Recent discoveries in the biology of ncRNAs, such as TUCRs, other lncRNAs, and miRNAs, have highlighted their roles as tumor suppressors and oncogenes. They have been causally linked to multiple cancers and can serve as biomarkers for diagnosis and disease prognosis and as targets for novel therapeutics [9].

In bladder cancer, uc.8 is the most upregulated TUCR. Uc.8 is a natural decoy for miR-596; thus uc.8 upregulation results in increased expression of MMP9, increasing the invasive potential of bladder cancer cells. Downregulation of uc.8 is associated with decreased cell accumulation, proliferation, invasion, and migration. Its expression is correlated with the grading and staging of patient tumors. Its localization serves as a potential biomarker; it is localized primarily to the cytosol in late-stage tumors, while it has both nuclear and cytosolic localizations in early-stage tumors [61,62]. In breast cancer, uc.51 is upregulated. It promotes proliferation and metastasis by stabilizing the non-POU domain-containing octamer-binding protein (NONO) [63]. SNPs in TUCRs such as uc.51 are associated with cancer risk [37].

TUCRs were found to have associations to prostate cancer development. Several TUCRs have also been responsive to epigenetic drugs or androgen. Experiments with human prostate cancer cell lines showed that uc.287 is induced by androgen whereas uc.283 was up-regulated following treatment with epigenetic drugs [47]. Upregulation of uc.338 is associated with poor prognosis in non-small cell lung cancer [64]. In liver, cervical, and lung cancer, uc.338 operates as an oncogene through the promotion of cell proliferation, migration, and invasion [65,66]. It accomplishes this effect through several potential targets, including as p21, CDK4, CDK6, Cyclin D1, Cyclin B, TIMP1, E2F1, MXD1, STIL, SH2D2A, BOP1, CCKBR, CYR61, DAB2, EGFR, EMP2, ERCC1, FOXC1, GFI1B, LAMA5, OSM, PES1, SHH, and ZEB2 [67].

In bladder and liver cancer, uc.306 is downregulated. Its role in the prognosis of hepatitis B (HBV)-related hepatocellular carcinoma (HCC) proves to be a promising biomarker. Uc.306 was obtained by screening microarray data obtained during the polarization of U937 cells from the M2 to M1 phenotype and was detected by qPCR. Uc.306 was upregulated in M1 cells and was predicted to be involved in the Wnt pathway. Low expression of uc.306 was significantly associated with a shorter overall survival [68]. In breast cancer, the lncRNA that is transcribed from uc.147 is associated with poor prognosis, despite being associated with the luminal A subtype, which typically has better prognosis. It is associated with the expression of miR-18 and miR-190b [69]. Expression of uc.73 in colorectal cancer is associated with positive outcomes; and yet, the directionality of its deregulation in this cancer is controversial. Knockdown of uc.73 in colorectal cells leads to a reduction in cell proliferation and an increase in cell apoptosis [53]. In gastric, colorectal, leukemia, and bladder cancers, uc.160 is upregulated. Inhibition of uc.160 reduces cell growth, viability, and proliferation. This could be a result of an indirect regulation of PTEN expression via a direct interaction with a serine/threonine protein kinase that is encoded by the AKT1 gene. It also has documented interactions with miR-24-1 and miR-155. Its expression signature can be used to distinguish leukemia from normal tissue and can even be used to identify disease subgroups [23,70,71].

Promoter methylation of the TUCRs uc.160, uc.283, and uc.346 has been reported in colorectal cancer. Notably, uc.160 and uc.283 methylation are associated with the grade of dysplasia in adenoma species. Higher uc.160 methylation was, predominantly in stage III and IV patients, linked to improved overall survival rate. Similarly, methylation of uc.283 is also associated with higher overall survival [23,70,71,72]. In colon cancer cell lines, overexpression of uc.160 and uc.346 led to increased proliferation and migration rates. Methylation levels of uc.160 in plasma of 50 CRC, 59 adenoma patients, 40 healthy subjects, and 12 patients with diverticulosis predicted the presence of CRC with 35% sensitivity and 89% specificity, while methylation levels of the combination of all three TUCRs resulted in 45% sensitivity and 74.3% specificity. In conclusion, TUCR expression and methylation status are deregulated in CRC while uc.160 and uc.346 appear to have a complex role in CRC progression. Moreover, the methylation status appears to be a promising non-invasive screening test for CRC, provided that assay sensitivity is improved [23].

Uc.216 was determined to be significantly associated with CpG oligodeoxynucleotide resistance in chronic lymphocytic leukemia, while uc.63 is associated with docetaxel resistance [73]. In prostate cancer cells, uc.308, uc.434, uc.241, uc.283, uc.285, uc.85, uc.287, uc.445, uc.134, and uc.240 are all upregulated by treatment with the epigenetic drugs 5-AzaC and TSA. Others, such as uc.249, uc.349, uc.204, uc.135, uc.31, uc.410, uc.344, and uc.283, were instead downregulated [47].

### 4.1. Caveats and Considerations in the Study of TUCRs

While TUCRs share many common traits with the superclass of lncRNAs, there are important distinctions and caveats to consider. Like other lncRNAs, TUCRs can be localized to the nucleus or the cytosol. This is important to consider when identifying potential mechanisms of action for a given TUCR. For example, we have summarized the effects of many TUCRs that interact with miRNAs. In the publications that performed fractionation experiments, miRNA-associated TUCRs were localized to the cytosol. This is consistent with their regulation of and by miRNAs, as the RISC machinery required for miRNA is commonly located in the cytosol [74].

It is useful to distinguish between the effects of TUCR RNAs and the UCR DNA regions from which they are transcribed [7]. Several studies have demonstrated that the TUCR sequences do not likely represent the full RNA transcript. These caveats are especially important to consider when considering intergenic TUCRs, which may represent novel lncRNAs themselves.

It is also important to consider the potential effects of UCR biology (DNA) on TUCR (RNA) biology. Many TUCRs overlap with ultraconserved enhancers, which have their own biology that may or may not be related to UCR or TUCR function [34,47,75,76]. TUCR sequences are often contained within other annotated genes. Even some intergenic TUCRs are only a few kilobases (kb) away from another annotated gene. While characterizing TUCR transcripts, it is critical to consider the expression and effects on host and neighboring genes [22].

While TUCRs are commonly considered to be lncRNAs, their definitive biological role has yet to be elucidated. TUCRs are poorly structured. This is in part due to the fact that the full transcripts and functional roles of most TUCRs remain unknown [3,9,11,14,41,74]. However, there are established methods for predicting and validating lncRNA structure that can be used when investigating TUCR biology. TUCR transcripts can be elucidated using methods such as 5′/3′ rapid amplification of cDNA ends (RACE) [54,77] or de novo RNA-Seq transcript reassembly integrated with TUCR annotations followed by PCR amplification [78]. TUCR structures can be identified using various computational and biological methods, such as fragmentation sequencing (frag-seq), dimethyl sulfate (DMS) probing, and selective 2′-hydroxyl acylation by primer extension (SHAPE) assays [79,80,81].

Identification of full TUCR transcripts may reveal genes that contain open reading frames (ORFs) and, therefore, coding potential. TUCR transcripts that are transcribed from enhancer regions may be instead acting as enhancer RNAs.

It has been suggested that TUCRs code for an organism’s essential functions. After identifying four TUCRs (uc.248, uc.329, uc.467, and uc.482) which would lead to profound phenotypic shifts when disturbed, knockout trains of mice which lacked one of the four elements were engineered. Results showed that the knockout mice survived and reproduced as expected, casting doubt on the prevailing notion that UCRs code essential genes [35,82]; and yet, a survey for UCRs was performed among the 26 cryptic genomic rearrangements detected in a series of 200 patients with idiopathic neurodevelopmental disorders. Pathogenic CNVs lacking UCRs showed almost a threefold higher content in genes. Presence in genomic imbalances of unknown effect might be suggestive of a clinically relevant condition. This suggests that enrichment of ultraconserved elements among genomic imbalances could affect mental delay and congenital anomalies, even if there is no reproductive or survival deficit [27,28,33].

For many TUCRs, a functional role has yet to be identified. Additionally, there is a lack information regarding the mechanism of action or tissue specificity of most TUCRs [3,9,11,14,41,74]. As a class, lncRNAs can either be ubiquitously expressed or tissue-specific. Their functions may vary not only by tissue context, but also by cellular localization [83,84,85]. Since TUCRs are generally considered to be lncRNAs, it is possible for TUCRs to have similar biology [3,9,11,14,41,74]. Significantly more research efforts are needed to better understand the roles of TUCRs in cancer and human disease.

### 4.2. Conclusions and Future Directions

TUCRs are 100% conserved in human, mouse, and rat genomes; 98% conserved in dog genomes; and 95% conserved in chicken genomes. This is remarkable, as lncRNAs are typically poorly conserved; and yet, the reason for this conservation remains unknown. Perhaps there is a shared motif among UCRs and TUCRs that explains this conservation.

We have summarized studies across several cancers, including neuroblastoma; breast, prostate, colorectal, bladder, liver, cervical, and lung cancer; leukemia; and esophageal squamous cell carcinoma (Table 2). As of this publication, there are no studies on TUCRs in many other cancers, including common cancers such as melanoma and pancreatic, uterine, and thyroid cancers. Greater insight is needed into the functional role that TUCRs play in these malignancies. Despite their potential clinical relevance, there is a limited understanding of how TUCRs can be leveraged as targets for cancer therapy. With few exceptions, TUCRs are mostly used as potential biomarkers for the initiation and progression of the investigated malignancies.

## Author Contributions

Conceptualization, M.K.G.J. and R.A.; Writing (Initial Draft), M.K.G.J., A.S., B.C. and C.R.-L.; Review and Editing, M.K.G.J., R.A., S.B., K.H., S.S., B.K., F.H., C.D. and Y.Z.; Funding Acquisition, R.A.; Project Supervision, R.A. All authors have read and agreed to the published version of the manuscript.

## Figures and Tables

**Figure 1 cells-11-01684-f001:**
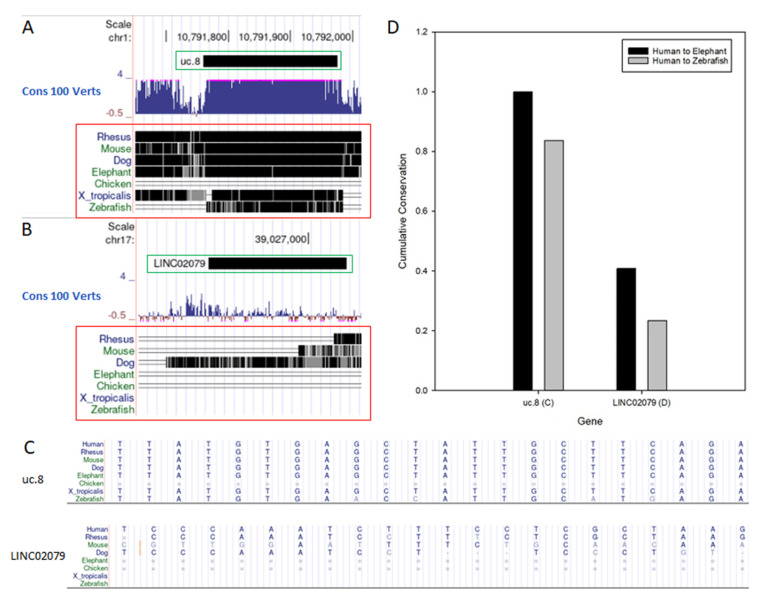
TUCRs represent highly conserved lncRNAs. LncRNAs are typically poorly conserved, while TUCRs are 100% conserved in human, mouse, and rat genomes. The TUCR uc.8 ((**A**), green box) is highly conserved ((**A**), red box) compared to the lncRNA LINC02079 ((**B**), green box), which is more poorly conserved ((**B**), red box). Nucleotide conservation is highlighted for uc.8 and LINC02079 (**C**) and quantified (**D**).

**Figure 2 cells-11-01684-f002:**
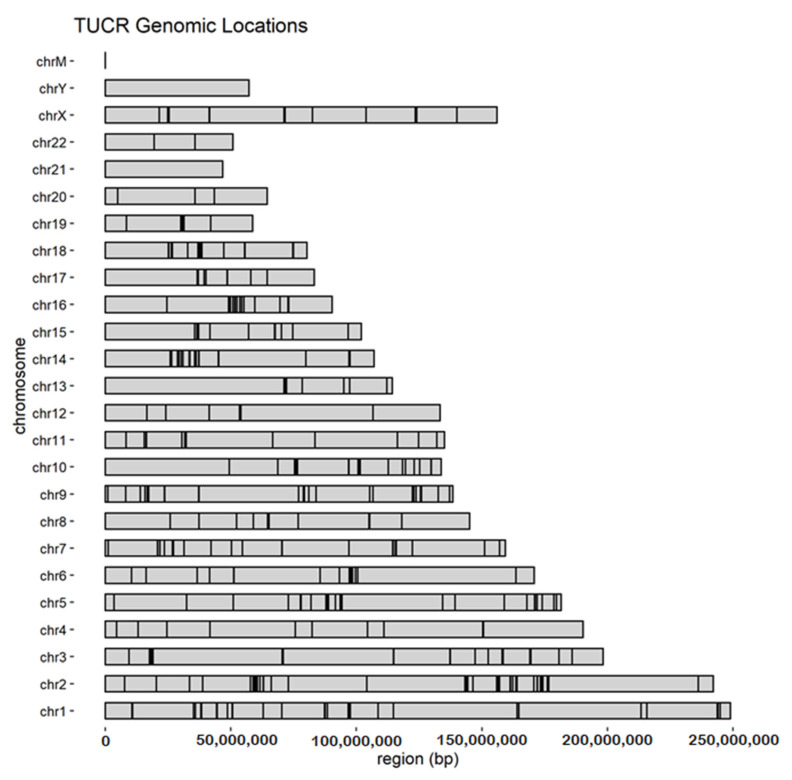
TUCRs are found on all number chromosomes and the X chromosome. TUCR locations are marked with a black vertical line showing their relative position on the chromosome. bp = base pair.

**Figure 3 cells-11-01684-f003:**
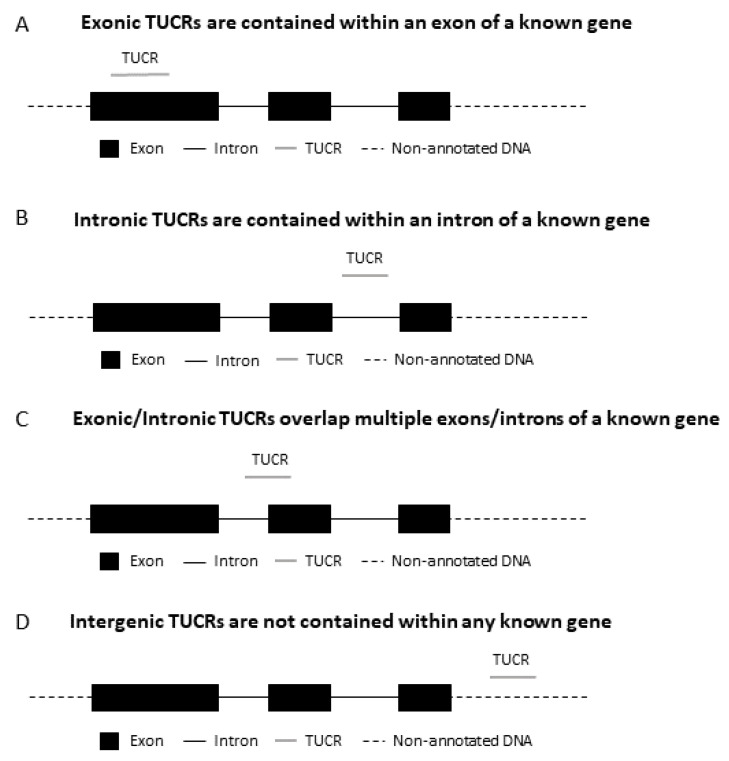
TUCRs can be intragenic or intergenic. Intragenic TUCRs can be found within the exon of their “host gene” (**A**), an intron (**B**), or straddling splice sites as exonic/intronic (**C**). The remainder exist outside of annotated DNA/RNA and are considered to be intergenic (**D**).

**Table 1 cells-11-01684-t001:** TUCRs are an understudied class of molecules. A PubMed search for the term “cancer” and “TUCR” or “ultraconserved” in manuscript abstracts reveals that they are understudied compared to lncRNAs, enhancer RNAs, antisense RNAs, miRNAs, and p53.

Primary Search Term	Secondary Search Term	Number of Publications
TUCR, UCR, or “ultraconserved”	Cancer	69
Long non-coding RNA	Cancer	7796
Enhancer RNA	Cancer	1207
Antisense RNA	Cancer	2310
Micro RNA	Cancer	12,141
p53	Cancer	24,073

**Table 2 cells-11-01684-t002:** A summary of known TUCR expression, functions, and mechanisms of action in cancer.

TUCR (Citation)	Cancer	Expression	Function	Mechanism
uc.8 [60,61,62]	Bladder	Upregulated	Invasion, Accumulation, Proliferation, Migration, Tumor Grading and Staging, Poor Prognosis	Upregulation of MMP9 via decoy for miR-596
uc.38 [50]	Breast	Downregulated	Reduced Proliferation, Induction of Apoptosis	BCL-2 family proteins through PBX-1 depletion
uc.51 [34,63]	Breast	Upregulated	Proliferation, Metastasis	Stabilization of NONO
uc.63 [43,44,45]	Breast, Prostate, Bladder, Chronic Lymphocytic Leukemia	Upregulated	Proliferation, Poor Prognosis, Docetaxel Resistance	Upregulation of MMP2 via miR-130b
uc.73 [53]	Colorectal	Upregulated/Downregulated	Cell Proliferation, Apoptosis	N/A
uc.83 [59]	Lung	Upregulated	Cell Growth	PI3K and MAPK signaling
uc.84 [60]	Breast	Upregulated	Cell Cycle Progression	Co-regulation w/miR-221 and CDKN1B expression
uc.110 [60]	Breast	Upregulated	Cell Cycle Progression	Co-regulation w/miR-221 and CDKN1B expression
uc.138 [46]	Colon	Upregulated	Cell Cycle Progression, Resistance to Apoptosis	N/A
uc.158 [57]	Hepatocellular Carcinoma	Upregulated	Cell Growth, Spheroid Formation, Migration, Apoptosis	Wnt Signaling and miR-193b
uc.160 [23,70,71,72]	Gastric, Colorectal, Leukemia, Bladder	Upregulated	Poor Prognosis, Proliferation, Migration, Promoter Methylation, Cell Growth	Indirect regulation of PTEN, Interaction w/miR-24-1 and -155
uc.183 [60]	Breast	Upregulated	Cell Cycle Progression	Co-regulation w/miR-221 and CDKN1B expression
uc.189 [51,52]	Esophageal Squamous Cell Carcinoma	Upregulated	Invasion, Advanced Disease, Metastasis, Poor Prognosis	N/A
uc.206 [58]	Cervical	Upregulated	Cell Growth	Direct regulation of p53
uc.216 [73]	Chronic Lymphocytic Leukemia	Upregulated	CpG oligodeoxynucleotide Resistance	N/A
uc.283 [47,56]	Prostate, Colorectal	Downregulated	Good Prognosis, Promoter Methylation	N/A
uc.306 [68]	Bladder, Liver	Downregulated	Good Prognosis	Wnt signaling
uc.338 [64,65,66,67]	Liver, Cervical, Lung	Upregulated	Proliferation, Migration, Invasion	Broad regulation of cell cycle genes
uc.339 [1,53,54,55]	Lung, Colorectal	Upregulated	Cell Proliferation, Clonogenic Growth, Soft Agar Growth, Adhesion	Competing RNA w/cyclin E2 for miR-339-3p, -663b-3p, and 95-5p
uc.346 [23,70,71,72]	Colorectal	Upregulated	Poor Prognosis, Proliferation, Migration, Promoter Methylation	N/A
uc.416 [49]	Renal Cell Carcinoma, Gastric	Upregulated	Cell Growth and Migration	Regulation of miR-153 and IGFBP6
uc.454 [1,47,48]	Non-Small Cell Lung Cancer, Bladder, Prostate	Downregulated	Reduced Proliferation, Induction of Apoptosis	HSPA12B or Ras signaling pathway
uc.475 [42]	Epithelial	Upregulated	Proliferation	N/A
uc.147 [69]	Breast	Upregulated	Poor Prognosis	Potential regulation of miR-18 and miR-190b

## Data Availability

This study does not report any data.

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
