# Peer review of "Transcribed Ultraconserved Regions in Cancer"

_cells, 2022, doi:10.3390/cells11101684_

Round 1
Reviewer 1 Report
I hesitate to recommend this paper for publication. It is not that the subject is not interesting, because it is, very much. But it is badly written and in my opinion poorly organized. Some sentences are poorly understandable: for instance (line 77), one reads: "TUCRs...maybe intragenic, or within an annotated genomic region, or intergenic.." which leaves the reader with the impression that there are three possible alternatives. It is only by referring to Fig.3 that one realizes that the alternatives are only two. There are several examples of such sloppy writing throughout the paper. Then, the ensuing paragraphs 3 and 4 are a long and rather disjointed list of instances in which UCRs have been found disregulated in many cancers, which leaves the reader rather confused as to the general significance of these observation, and with no clear take-home message. I think that the paper requires extensive rewriting and re-planning before it can become really useful for the average reader.
Author Response
Reviewer 1 Comments:
I hesitate to recommend this paper for publication. It is not that the subject is not interesting, because it is, very much. But it is badly written and in my opinion poorly organized. Some sentences are poorly understandable: for instance (line 77), one reads: "TUCRs...maybe intragenic, or within an annotated genomic region, or intergenic.." which leaves the reader with the impression that there are three possible alternatives. It is only by referring to Fig.3 that one realizes that the alternatives are only two. There are several examples of such sloppy writing throughout the paper. Then, the ensuing paragraphs 3 and 4 are a long and rather disjointed list of instances in which UCRs have been found disregulated in many cancers, which leaves the reader rather confused as to the general significance of these observation, and with no clear take-home message. I think that the paper requires extensive rewriting and re-planning before it can become really useful for the average reader.
Response:
We appreciate these constructive comments from the reviewer. We have edited the line in question (now line 90) for clarity. We have also made extensive changes in the formatting and organization of the manuscript.
Reviewer 2 Report
Recent discoveries facilitated by next-generation sequencing technologies have revealed multiple roles of RNA beyond its conventional roles in protein synthesis. These non-coding RNAs play critical roles in almost all expects of gene expression regulation, modification of other RNAs, and formation of functional complexes with proteins and metabolites. Thus, many non-coding RNAs are implicated in many diseases, including various cancers. In this review, Gibert Jr. et al. have presented a comprehensive review of a relatively new but exciting class of non-coding RNAs – TUCRs. Authors have discussed different aspects of these noncoding RNAs and how they are linked with various diseases such as cancers.
The manuscript is well-written, comprehensive, and not difficult to follow. I think this review would be helpful for the ncRNA research community and beyond; therefore, the manuscript is worth publishing in a reputed journal like Cells. However, I have some comments that I expect authors would address before the manuscript could be accepted for publication.
1. The manuscript needs more careful proofreading for typos and grammatical errors. For example, figure 1 legend ends with two periods.
2. The quality of the figures needs to be improved in size and resolution (especially figure 1).
3. In figure 1, it would be helpful to show the actual sequence alignments to demonstrate the high or low conservation. What does it mean by more poor conservation – in terms of percentage?
4. Authors discussed that “a PubMed literature search reveals less than 75 papers that contain the words T-UCR, TUCR, or ultraconserved and the words cancer, malignancy, and/or tumorigenesis. Compare this to a single gene, such as TP53 (p53), which is contained thousands of publications under the same search parameters.”
It would be nice to see this data in some sort of graphical presentation. Also, a comparison with another class of non-coding RNA would give an excellent perspective.
5. Worth discussing the structural view of TUCRs briefly – are these RNAs well-structured?
Author Response
Reviewer 2 Comments:
Recent discoveries facilitated by next-generation sequencing technologies have revealed multiple roles of RNA beyond its conventional roles in protein synthesis. These non-coding RNAs play critical roles in almost all expects of gene expression regulation, modification of other RNAs, and formation of functional complexes with proteins and metabolites. Thus, many non-coding RNAs are implicated in many diseases, including various cancers. In this review, Gibert Jr. et al. have presented a comprehensive review of a relatively new but exciting class of non-coding RNAs – TUCRs. Authors have discussed different aspects of these noncoding RNAs and how they are linked with various diseases such as cancers.
The manuscript is well-written, comprehensive, and not difficult to follow. I think this review would be helpful for the ncRNA research community and beyond; therefore, the manuscript is worth publishing in a reputed journal like Cells. However, I have some comments that I expect authors would address before the manuscript could be accepted for publication.
Response:
We appreciate these constructive and considerate feedback regarding our manuscript submission. We have attempted to address any comments and/or concerns with the solutions outlined below.
- The manuscript needs more careful proofreading for typos and grammatical errors. For example, figure 1 legend ends with two periods.
We have carefully reviewed the manuscript for typos and grammatical errors, and have corrected several that were missed during the initial submission.
- The quality of the figures needs to be improved in size and resolution (especially figure 1).
We have increased the size of all figures, and attempted to improve the resolution as well. Additional panels have been added, primarily to answer the following question.
- In figure 1, it would be helpful to show the actual sequence alignments to demonstrate the high or low conservation. What does it mean by more poor conservation – in terms of percentage?
We have included an additional panel that highlights nucleotide conservation in uc.8 (a TUCR) and a comparatively sized lncRNA (LINC02079). We have also quantified this highlighted region, showing that a typical lncRNA is much less conserved than a TUCR, which are 100% conserved in humans, mice, and rats by definition. Additional text was added to the manuscript (lines 40-43) for clarity.
- Authors discussed that “a PubMed literature search reveals less than 75 papers that contain the words T-UCR, TUCR, or ultraconserved and the words cancer, malignancy, and/or tumorigenesis. Compare this to a single gene, such as TP53 (p53), which is contained thousands of publications under the same search parameters.”
It would be nice to see this data in some sort of graphical presentation. Also, a comparison with another class of non-coding RNA would give an excellent perspective.
We have introduced a new table (now Table 1) which quantifies the disparity in publications for TUCRs when compared to lncRNAs, antisense RNAs, enhancer RNAs, miRNAs, and p53.
- Worth discussing the structural view of TUCRs briefly – are these RNAs well-structured?
Many TUCRs are contained within larger RNA transcripts, of which the ultraconserved region is just a fraction. Because of this, the structural biology of TUCRs as a class is relatively understudied, with only a few publications investigating the structure of individual TUCRs. However, there are existing methods for investigating RNA structure. We have added additional text to the discussion to further elaborate on this question (lines 328-337)
Reviewer 3 Report
This is a well written review providing extensive coverage of an interesting topic, ultra conserved regions. The authors could consider addressing a few questions that I had after reading the review.
- It looks like UCRs are non-randomly distributed in the genome. I think that they are enriched at fragile sites, but are there any other thoughts on how they came to be enriched where they are? (Figure 2)
- Do the SNPs discussed in section 2 have any known function?
- How can UCRs be enriched in fragile sites and be highly conserved? Do they not get regularly mutated?
- Could the nomenclature for UCRs be clarified. I get why a UCR in Chromosome 1 might be called UC.1, but why would a UCR in the x chromsome be called uc.481?
- It seems odd to discuss UC206's association with p53 expression in cervical cancer given that p53 is essentially non-functional in cervical cancer due to HPV mediated degradation. Is the decrease biologically relevant?
- Some of the UCRs discussed are associated with major changes in tumor suppressor activity. Are these (or any others) more likely to cause cancer in multiple tissue types, like p53 inactivation is associated with cancer development in a non-tissue specific manner.
Author Response
Reviewer 3 Comments:
This is a well written review providing extensive coverage of an interesting topic, ultraconserved regions. The authors could consider addressing a few questions that I had after reading the review.
Response:
We are grateful for the generous and constructive feedback provided by this reviewer. We have attempted to address each of their comments/suggestions below:
1. It looks like UCRs are non-randomly distributed in the genome. I think that they are enriched at fragile sites, but are there any other thoughts on how they came to be enriched where they are? (Figure 2)
As of right now, there is a limited understanding of TUCR biology, including why they are ultraconserved. We have not yet been able to identify an explanation in the literature for why TUCRs arise at their genomic loci. However, there are some hypotheses, which we have now included in the manuscript (lines 53-59)
2. Do the SNPs discussed in section 2 have any known function?
One of the SNPs has a predicted effect on protein/peptide structure, while another is identified as an intron variant and may or may not be less deleterious. The remaining SNPs that are discussed in this manuscript have only been investigated in terms of their relationship to TUCRs. These SNPs are associated with disease risk, but the functional and/or causal role has yet to be investigated.
We have added some text to the appropriate section for clarity (lines 156-168)
3. How can UCRs be enriched in fragile sites and be highly conserved? Do they not get regularly mutated?
Perhaps counterintuitively, TUCRs are enriched in fragile sites, and remain conserved despite an expectation that they would be subject to frequent mutation. This is an area of research that requires further investigation. We have added additional text to the manuscript to reflect this caveat. (lines 124-125)
4. Could the nomenclature for UCRs be clarified. I get why a UCR in Chromosome 1 might be called UC.1, but why would a UCR in the x chromsome be called uc.481?
For the purpose of TUCR nomenclature, the X chromosome is considered chromosome 23, Y is considered chromosome 24. We have added additional text to the manuscript for clarity. (line 63)
5. It seems odd to discuss UC206's association with p53 expression in cervical cancer given that p53 is essentially non-functional in cervical cancer due to HPV mediated degradation. Is the decrease biologically relevant?
This is a particularly remarkable comment, one that required a return to the manuscript in question. This functional role was identified in cervical cancer cell lines, perhaps in the absence of HPV. This merits further insight into the role of uc.206 in the initiation and progression cervical cancer within the context of HPV. Additional text has been added to the manuscript to reflect this important caveat. (lines 239-242)
6. Some of the UCRs discussed are associated with major changes in tumor suppressor activity. Are these (or any others) more likely to cause cancer in multiple tissue types, like p53 inactivation is associated with cancer development in a non-tissue specific manner.
It is entirely possible for some TUCRs to have tissue specific effects, as the localization and function of TUCRs may vary depending on the tissue. However, TUCR that have the same localization and function in different tissues may be operating in a more generic manner. Additional text has been added to the discussion to address the tissue specificity of TUCRs. (360-366)
Reviewer 4 Report
The manuscript attempts to collate interesting information regarding the relationship between tumor and Transcribed ultra-conserved regions (TUCR) in our genome. The role of TUCR in tumor, as suggested by the authors, is a quite an understudied topic. Despite the potential clinical relevance of TUCR, the knowledge is limited about how TUCRs can be leveraged as targets for cancer therapy. Hence, the information and the viewpoint about TUCR put forth by the review is important and scientifically valid. This is a well written review with substantial clinical relevance and worth publishing in this journal.
However, I have a concern regarding the title. The term “mysterious”, which could be avoided from the title since it did not sound scientific in the context of a review. Please use an alternative title.
Author Response
Reviewer 4 Comments:
The manuscript attempts to collate interesting information regarding the relationship between tumor and Transcribed ultra-conserved regions (TUCR) in our genome. The role of TUCR in tumor, as suggested by the authors, is a quite an understudied topic. Despite the potential clinical relevance of TUCR, the knowledge is limited about how TUCRs can be leveraged as targets for cancer therapy. Hence, the information and the viewpoint about TUCR put forth by the review is important and scientifically valid. This is a well written review with substantial clinical relevance and worth publishing in this journal.
However, I have a concern regarding the title. The term “mysterious”, which could be avoided from the title since it did not sound scientific in the context of a review. Please use an alternative title.
Response:
We thank the review for their generous comments and suggestions and have removed the term “mysterious” from the title of the manuscript.
Reviewer 5 Report
This an excellent review about Mysterious Transcribed Ultraconserved Regions in Cancer. It covers an important area which has not been reviewed before. Overall the review is sufficiently infomative and appears to be well written.
The reviewer feels that the manuscript is worthy enough to be published in this journal.
Author Response
Reviewer 5 Comments:
This an excellent review about Mysterious Transcribed Ultraconserved Regions in Cancer. It covers an important area which has not been reviewed before. Overall the review is sufficiently informative and appears to be well written.
The reviewer feels that the manuscript is worthy enough to be published in this journal.
Response:
We appreciate these generous comments regarding the submission of this review on TUCRs in Cancer.
Round 2
Reviewer 1 Report
The revised version of the paper is only marginally improved with respect to the previous one. However, it contains information that may be useful for investigators in the field, so it may be published.